# Proposal of Four New *Aureobasidium* Species for Exopolysaccharide Production

**DOI:** 10.3390/jof9040447

**Published:** 2023-04-06

**Authors:** Feng Wu, Zixuan Feng, Manman Wang, Qiming Wang

**Affiliations:** 1School of Life Sciences, Institute of Life Sciences and Green Development, Hebei University, Baoding 071002, China; 2Engineering Laboratory of Microbial Breeding and Preservation of Hebei Province, Hebei University, Baoding 071002, China; 3Key Laboratory of Microbial Diversity Research and Application of Hebei Province, Hebei University, Baoding 071002, China

**Keywords:** black yeasts, four new species, molecular phylogeny

## Abstract

In this study, 99 strains of *Aureobasidium* species were isolated from various samples collected from different locations in China, among which 14 isolates showed different morphological characteristics to other strains identified as known *Aureobasidium* species. Based on morphological characteristics, those 14 strains were classified into four groups, represented by stains of KCL139, MDSC−10, XZY411−4, and MQL9−100, respectively. Molecular analysis of the internal transcriptional spacer (ITS) and part of the large ribosome subunit (D1/D2 domains) indicated that those four groups represent four new species in the *Aureobasidium*. Therefore, the names *Aureobasidium insectorum* sp. nov., *A. planticola* sp. nov., *A. motuoense* sp. nov., and *A. intercalariosporum* sp. nov. are proposed for KCL139, MDSC−10, XZY411−4, and MQL9−100, respectively. We also found that there were differences in the yield of exopolysaccharides (EPS) among and within species, indicating strain-related exopolysaccharide-producing diversity.

## 1. Introduction

*Aureobasidium* (Ascomycota: Dothideales) is a yeast-like fungal genus that is often called black yeast because of the production of melanin during its growth [1,2,3]. Species of *Aureobasidium* are widely distributed and normally possess multiple trophic modes. They are often found as saprophytes, endophytes, and pathogens in diverse environments, such as plant materials (roots, leaves, bark), water, marine sediments, swamps, soil, air, skin, and high osmotic environments (significant osmotic stress) [4,5,6,7,8,9,10,11]. The species of *Aureobasidium* produce one-celled conidia of various shapes from hyaline and terminal, lateral, or intercalary conidiogenous cells [4,9].

The genus *Aureobasidium* was first described by Viala and Boyer based on the isolates on grape leaves [10]. Hermanides-Nijhof assessed the phenotypic variety of *Aureobasidium* and related genera, and distinguished *Aureobasidium* from the related enteric *Hormonema* according to the mode of conidial production. *Aureobasidium* produced synchronous blastoconidia from undifferentiated, hyaline cells, whereas *Hormonema* produced conidia in basipetal succession from hyaline or dark hyphae [4]. De Hoog and Yurlova revised *Aureobasidium* taxonomy based on the morphology, physiology, and biochemistry, and thus the genus included three species: *Aureobasidium pullulans*, *A. melanogenum,* and *A. aubasidani* [5,6,7]. In 2008, Zalar et al. carried out a molecular analysis of *A. pullulans* and *A. melanogenum,* and identified the studied strains as *A. pullulans*, *A. melanogenum*, *A. subglaciale*, and *A. namibiae* [8,9]. In recent years, with the high accessibility of sequencing services and a large amount of available molecular data, the number of novel *Aureobasidium* species is increasing. Thirteen new species have been proposed, namely, *Aureobasidium acericola* [10], *A. aerium* [11], *A. castaneae* [12], *A. iranianum* [13], *A. leucospermi* [14], *A. mangrovei* [15], *A. microtermitis* [16], *A. mustum* [17], *A. pini* [18], *A. thailandense* [19], *A. tremulum* [20], and *A. uvarum* and *A. vineae* [17].

With the increasing number of *Aureobasidium* species, its functional activities have been explored. For example, *Aureobasidium* species are resistant to *Botrytis cinerea* and *Rhizopus stolonifera* as biological control agents, and they can also be used as sources of single-cell proteins [21,22]. *Aureobasidium pullulans* is often fermented to produce β-polymalic acid, laccase, liamocins, pullulan polysaccharides, and other commercial compounds [14,23,24,25,26,27,28]. Liamocins display pharmacological activities including anti-*Streptococcus* and anticancer [29]. Pullulan polysaccharides are non-toxic, tasteless, harmless, degradable, water-soluble, stable, film-forming, and present other excellent properties. As raw materials for food and cosmetics, pullulan polysaccharides are used for immune regulation, anti-tumor and anti-metastasis, relief of influenza and food allergy, and relief of stress [30].

Pullulan polysaccharides produced by *A. pullulans* have the properties of water retention, barrier formation, regeneration, whitening, hydrating, and repairing, and are added as an ingredient to cosmetic formulas. In order to screen an excellent *A. pullulans* to make a fermentation of excellent efficacy for cosmetics, we collected various samples from Tibet, Zhejiang, Yunnan, Shaanxi, Gansu, Hebei, and other areas in China for strain isolation. Ninety-nine strains belonging to *Aureobasidium* were obtained, among which 14 strains were identified as four new species, based on morphological characteristics and molecular analysis of the internal transcriptional spacer (ITS) and part of the large ribosome subunit (D1/D2 domains).

## 2. Materials and Methods

### 2.1. Sample Collection and Strain Isolation

Two hundred leaf samples of Sea buckthorn, willow, oak, crabapple, privet, camphor, and poplar collected from Motuo County in Tibet, and the Yunnan, Shaanxi, Gansu, and Zhoushan regions in Zhejiang Province, were cut into small pieces under sterile conditions and soaked in 0.1% Tween 80. The Tween solution from the leaves was diluted 10 times and directly spread on a potato dextrose agar medium (PDA; 200 g potatoes, 20 g glucose, 20 g agar per L) containing chloramphenicol (0.1 mg/mL) and cultured at 28 °C for 48 h.

Ten insect samples were collected from Xiaowutai Mountain, Zhangjiakou region, Hebei Province. Insect samples with 75% alcohol disinfection were removed from the head and ground with a grinder, then soaked in 0.1% Tween 80. The Tween solution from the inset was diluted 10 times and directly spread on PDA containing chloramphenicol (0.1 mg/mL) and cultured at 28 °C for 48 h.

### 2.2. DNA Isolation, PCR Amplification and Sequencing

DNA was extracted using the CTAB method [31]. ITS region of ribosomal DNA and the D1/D2 domains of the ribosome subunit (LSU) were amplified and sequenced with the primer pairs of ITS1/ITS4 (ITS1 5′ —GTC GTA ACA AGG TTT CCG TAG GTG— 3′; ITS4 5′ —TCC TCC GCT TAT TGA TAT GC— 3′) and NL1/NL4 (NL1 5′ —GCA TAT CAA TAA GCG GAG GAA AAG— 3′; NL4 5′ —GGT CCG TGT TTC AAG ACG G— 3′) [32,33].

The PCR reaction was performed in the 25 µL reaction mixture containing 0.5 µL of each primer (10 pM/µL), 1.0 µL of genomic DNA (10 ng/µL), and 23 µL of 1 × PCR Master Mix buffer (T3 Super PCR Mix, 10 × 1.125 mL, Tsingke Biotechnology Co., Ltd., Beijing, China). Amplification was performed in an AB 2720 thermal cycler (Applied Biosystems, Foster City, California, USA), with the program consisting of 98 °C for 2 min, 35 cycles of 98 °C for 10 s, 52 °C for 10 s, and 72 °C for 15 s, and the last elongation at 72 °C for 5 min.

### 2.3. Observation of Morphology

The isolates were cultured on PDA, oatmeal agar (OA; 30 g oatmeal, 20 g agar per L), and malt extract agar (MEA; 50 g malt extract, 20 g agar per L), and incubated in darkness at 25 °C for one week, in order to obtain morphological descriptions, including colony color and appearance. Fungal structures were transferred to microscope slides and mounted on 85% lactic acid drops. M40Y and M60Y media were prepared as described [15,19]. MEA media containing salt were prepared by adding analytical grade NaCl to the MEA prior to sterilization. Cultures were incubated at 25 °C unless otherwise noted.

The microscopes are equipped with LEICA DM2500 cameras (LECIA, Wetzlar, Germany) and use LASV4.13 software. At least 50 representative measurements were randomly selected and measured to calculate the average size.

### 2.4. Phylogenetic Analyses

For phylogenetic analyses, 28 new ITS and LSU sequences were obtained from the present study, and 56 reference sequences from GenBank (Table 1).

ITS and D1/D2 sequences were aligned with the Muscle program in MEGA7 [34], and minor gaps in all alignments were manually deleted. The most appropriate model of DNA substitution was searched with MEGA7 [35]. The model GTR + I + G was selected for Maximum likelihood (ML) and Bayesian inference (BI) analyses. ML analysis was carried out using MEGA7 [35] with 1000 bootstrap replicates. Bayesian inference (BI) analysis was conducted using MrBayes 3.1.2 [34] with 10,000,000 generations, and parameter settings were proposed by Wang et al. [36]. The phylogenetic tree and the alignments were deposited in TreeBASE (www.treebase.org, accessed on 21 December 2022, No. 30010).

### 2.5. Exopolysaccharides Production

The starter cultures of the 29 strains (Table 2) were prepared by cultivating the strains in 50 mL of inoculum medium containing 10 g yeast extract, 20 g peptone, and 20 g glucose per liter of distilled water at 25 °C for 3 days on a 150-rpm rotary shaker. Then, 5 mL of the starter culture was transferred to 100 mL synthetic medium containing 100 g sucrose, 1.7 g yeast extract, 5 g K_2_HPO_4_, 0.2 g MgSO_4_.7H_2_O, 0.6 g (NH_4_)_2_SO_4_ and 1.0 g NaCl per liter of deionized water at 28 °C for 6 days on a 150-rpm rotary shaker, the culture obtained was centrifuged at 7104 × g for 5 min, and the supernatant was collected. In order to precipitate exopolysaccharides, the cold ethanol was added to the obtained solution at a volumetric ratio of 2:1 *v*/*v* and the resulting mixture was kept in a refrigerator at 4 °C for 16 h [37]. The cold mixture was further centrifuged at 7104× *g* at 4 °C for 15 min and the pellet was dried overnight at 45 °C and then weighed to determine the yield of crude EPS production [38].

**Table 1 jof-09-00447-t001:** Names, strains, locations, and hosts, and corresponding GenBank numbers of the taxa used in this study.

Species	Strain	Date	Location	Latitude and Longitude	Source	GenBank No.		References
						ITS	D1/D2	
*Aureobasidium acericola*	CDH 2020−10	June 2020	South Korea	37°45′49.50″ N, 127°11′3.8″ E	*Acer pseudosieboldianum*	MT863788	MT863787	[10]
*Aureobasidium aerium*	CFCC 50324	April 2015	Sennon, Beijing, China	NA	air	ON007058	ON007081	[11]
*Aureobasidium castanea*	CFCC 54591 *	November 2021	Jinjing Town, Changsha Hunan, China	28°58′52″ N, 113°34′38″ E	*Castanea heryi*	NR_177551	MW364275	[12]
*Aureobasidium caulivorum*	CBS 242.64	NA	Oregon, America	NA	*Trifolium incarnatum*	FJ150871	FJ150944	[39]
***Aureobasidium insectorum* sp. nov.**	**KCL139**	**September 2021**	**Zhangjiakou, Hebei, China**	**39°30′ N, 113°50′ E**	**spittle insects**	**OP856707**	**OP857208**	**This study**
	**LPL−1C**	**September 2022**	**Zhoushan, Zhejiang, China**	**29°53′28.86″ N, 122°24′59.35″ E**	**leaf**	**OP856705**	**OP857207**	**This study**
	**XZY65−10**	**October 2019**	**Shannan City, Tibet, China**	**29°14′9.68** **″ N, 91°45′59.50″ E**	**leaf**	**OP856706**	**OP857206**	**This study**
	**L2PL−7A**	**September 2022**	**Zhoushan, Zhejiang, China**	**29°53′28.86″ N, 122°24′59.35″ E**	**leaf**	**OP856715**	**OP857216**	**This study**
	**T1−27−2**	**November 2021**	**Motuo County, Tibet, China**	**29°19′37.128″ N, 95°19′53.76″ E**	**leaf**	**OP856714**	**OP857215**	**This study**
	**XZY249M1**	**October 2019**	**Nyingchi City, Tibet, China**	**29°19′37.128″ N, 95°19′53.76″ E**	**deadwood**	**OP856713**	**OP857214**	**This study**
	**XZY63−10**	**October 2019**	**Shannan City, Tibet, China**	**29°14′9.68″ N, 91°45′59.50″ E**	**leaf**	**OP856712**	**OP857213**	**This study**
***Aureobasidium intercalariosporum* sp. nov.**	**MGL11−3**	**September 2022**	**Zhoushan, Zhejiang, China**	**29°53′28.86″ N, 122°24′59.35″ E**	**leaf**	**OP856703**	**OP857204**	**This study**
	**MQL9−100**	**September 2022**	**Zhoushan, Zhejiang, China**	**29°53′28.86″ N, 122°24′59.35″ E**	**leaf**	**OP856703**	**OP857205**	**This study**
*Aureobasidium iranianum*	CCTU 268	June 2009	Southern parts of Iran	NA	bamboo stems	NR_137598	NG_057049	[13]
*Aureobasidium khasianum*	NFCCI 4275	December 2016	Meghalaya, India	NA	litter samples	MH188305	MH188306	[40]
*Aureobasidium leucospermi*	CBS 130593	April 2008	South Africa	NA	leaves and stems of Proteaceae with cankers or leaf spots	NR_156246	MH877257	[14]
*Aureobasidium lini*	CBS 125.21T	NA	UK	NA	*Linum usitatissimum*	FJ150897	FJ150946	[8]
*Aureobasidium mangrovei*	IBRCM 30265T	January 2016	Qeshm Island, Iran	26°47′ N, 55°45′ E	mangrove trees (*Avicennia marina*)	NR_174637	NG_078639	[15]
*Aureobasidium melanogenum*	CBS 105.22	NA	NA	NA	**leaf**	**NR_159598**	**NG_056960**	**[8]**
*Aureobasidium microstictum*	CBS 342.66	NA	Germany	NA	dying or dead leaves	KT693743	FJ150945	[8]
*Aureobasidium microstictum*	CBS 114.64	NA	Wageningen, The Netherlands	NA	*Hemerocallis* sp.	KT693744	KT693986	[8]
*Aureobasidium microtermitis*	NA	NA	NA	NA	NA	MW276135	MW276136	NA
***Aureobasidium motuoense* sp. nov.**	**E82−2**	**October 2019**	**Motuo County, Tibet, China**	**29°19′37.128″ N, 95°19′53.76″ E**	**soil**	**OP856702**	**OP857203**	**This study**
	**XZY411−4**	**August 2019**	**Motuo County, Tibet, China**	**29°19′37.128″ N, 95°19′53.76″ E**	**leaf**	**OP856710**	**OP857211**	**This study**
	**E31−1**	**October 2019**	**Motuo County, Tibet, China**	**29°19′37.128″ N, 95°19′53.76″ E**	**soil**	**OP856709**	**OP857210**	**This study**
	**E26−4**	**October 2019**	**Motuo County, Tibet, China**	**29°19′37.128″ N, 95°19′53.76″ E**	**soil**	**OP856708**	**OP857209**	**This study**
*Aureobasidium mustum*	AWRI 4233 CO−2020	NA	South Australia	NA	grape juice	NA	NA	[17]
*Aureobasidium namibiae*	CBS 147.97	1997	Namib Desert, Namibia	NA	dolomitic marble	FJ150875	FJ150937	[8]
*Aureobasidium pini*	CFCC 52778	May 2018	Miyun District, Beijing, China	40°41′18″ N, 116°55′21″ E	pine needles covered with mycelium	MK184533	MK184535	[18]
***Aureobasidium planticola* sp. nov.**	**MDSC−10**	**September 2022**	**Zhoushan, Zhejiang, China**	**29°53′28.86″ N, 122°24′59.35″ E**	**leaf**	**OP856711**	**OP857212**	**This study**
*Aureobasidium proteae*	CBS 114273	February 2006	Netherlands	NA	*Protea* sp.	JN712491	JN712557	[15]
*Aureobasidium proteae*	CPC 13701	July 1998	Hilly Lands Farm, Somerset West, South Africa	NA	*Protea* cv. ‘*Sylvia*’	JN712490	JN712556	[15]
*Aureobasidium pullulans*	CBS 584.75	1974	France	NA	fruit of *Vitis vinifera*	FJ150906	FJ150942	[8]
*Aureobasidium pullulans*	CBS 146.30	NA	Germany, Ohlsdorf near Hamburg	NA	*slime flux of Quercus* sp.	FJ150902	FJ150916	[8]
*Aureobasidium subglaciale*	EXF−2481	June and August 2001	Norway, Svalbard, Kongsvegen	79° N, 12° E	subglacial ice from seawater	FJ150895	FJ150913	[8]
*Aureobasidium thailandense*	NRRL 58539T	2006	Nakhonratchasima, Thailand	NA	leaf of *Cerbera odollum*	JX462674	JX462674	[19]
*Aureobasidium thailandense*	NRRL 58543	2006	Prachuapkhirikhan, Thailand	NA	wood surface	JX462675	JX462675	[19]
*Aureobasidium tremulum*	UN 1	NA	NA	NA	NA	MK503657	MK503660	NA
*Aureobasidium uvarum*	AWRI 4620 CO−2020	NA	NA	NA	NA	NA	NA	[17]
*Aureobasidium vineae*	AWRI4619 CO−2020	NA	NA	NA	NA	NA	NA	[17]
*Selenophoma mahoniae*	CBS 388.92	NA	Colorado, America	NA	*Mahonia repens*, leaf	FJ150872	FJ150943	[8]
*Sydowia polyspora*	CBS 750.71	September 1969	Quebec, Lac Normand, Canada	NA	*Pinus strobus*, twig	MH872085	MH872085	[41]

* Note: Generated sequences and new strains in this study are indicated in bold. NA: Not available.

## 3. Results

### 3.1. Phylogeny

The phylogenetic tree, based on a combined dataset of the ITS region and D1/D2 domain of the LSU sequences, was used to resolve the taxonomic position of the newly collected strains within *Aureobasidium*.

Fourteen newly isolated strains in this study were formed into four separate groups (Figure 1). Strains E82−2, XZY411−4, E31−1, and E26−4 were closely related to *A. acericola*, *A. melanogenum* and *A. mustum*, with 81% bootstrap support. Strains KCL139, LPL−1C, XZY65−10, L2PL−7A, T1−27−2, XZY249M1, and XZY63−10 formed a basal clade related to *A. acericola*, *A. melanogenum*, *A. mustum*, *A. uvarum*, *A. vineae*, *A. aerium*, *A. subglaciale*, *A. leucospermi,* and *A. khasianum*, but without support. MGL11−3 and MQL9−100 clustered together with a separate clade. Strain MDSC−10 located at a basal branch related to *A. thailandense*, *A. microtermitis*, and *A. castaneae*.

### 3.2. Taxonomy

#### 3.2.1. *Aureobasidium insectorum* Q.M. Wang, F. Wu & M.M. Wang sp. nov.

Fungal names no: FN 571251

Etymology: Referring to the insect cicada, where the type of strain originated.

Colonies grew moderately on PDA, MEA, and OA (Figure 2), attaining 34 mm, 34 mm, and 27 mm diameters after 7 days of incubation at 25 °C, respectively. Colonies on PDA were flat, smooth, pitch black with white fimbriate margins, and lacking aerial mycelium. Colonies on MEA were flat, felty, and greenish-black with white fimbriate margins. Colonies on OA were flat, whitish, and olivaceous black in the center, with sparse aerial mycelium. The growth diameters, sugar and salt tolerance, and different cardinal growth temperature of *A. insectorum* are shown in Table 3. *A. insectorum* can grow at 4–30 °C, and the optimum growth temperature is 28 °C. On MEA supplemented with 15% (*w*/*v*) NaCl, the diameters of *A. insectorum* attained 8–9 mm, while *A. mangrovei* and *A. pullulans* were 5 mm and 7 mm, respectively [10,19]. Thus, *A. insectorum* grew stronger on concentrations of 15% NaCl than its closely related species. *A. insectorum* grew stronger on moderate-sugar-level media (PDA) and M40Y, and weaker on M60Y. This species is suitable to grow on M40Y and PDA.

Mycelium hyaline were smooth, thin-walled, and 5.8–10.6 µm (av. = 8.0 µm) wide. Conidiophores were not developed. Conidiogenus cells were holoblastic, smooth, cylindrical, and 11.65–4.59 µm (av. = 7.16 µm) wide. Conidia were hyaline, aseptate, smooth walled, ellipsoidal to elongate-ellipsoidal, straight or slightly curved, 4.2–7.4 × 1.7–3.5 μm (av. = 5.2 × 2.8 μm), often polar or bipolar buds. Chlamydospore were 3.5–8.9 × 1.3–6.7 μm (av. = 6.5 × 5.4 μm), black, smooth, globose to ellipsoidal, septate or aseptate, and constricted at the septa (Figure 3).

Material examined: —CHINA, Hebei, Zhangjiakou, Xiaowutai Mountain, from cicada, L. Min, August 2021. (Holotype HMAS 352303; ex-type culture CGMCC 2.7207 = KCL139).

Notes—This species is phylogenetically close to *A. aerium*, *A. intercalariosporum*, *A. leucospermi,* and *A. khasianum*. These five species can be distinguished by the sizes of conidia (4.2–7.4 × 1.7–3.5 μm in *A. insectorum*, vs. 12.8–19.5 × 7.9–11.9 μm in *A. aerium*, vs. 8–13 × 5–9/8–24 × 2–10 μm in *A. leucosperm*, vs. 3–4 × 2–40 μm in *A. khasianum*, vs. 10.5–17.1 × 10.5–12.9 μm in *A. intercalariosporum*) [11,12,40].

#### 3.2.2. *Aureobasidium planticola* Q.M. Wang, F. Wu & M.M. Wang sp. nov.

Fungal names no: FN 571262.

Etymology: Referring to the plant where the ex-type strain originated.

Colonies grew moderately on PDA, MEA, and OA (Figure 2), attaining 22 mm, 24 mm, and 30 mm diameters after 7 days of incubation at 25 °C, respectively. Colonies on PDA were flat, pitch black, and pale grey in the center. Colonies on MEA were flat, felty, and brown, with white fimbriate at the margin. Colonies on OA had a smooth margin, and were flat, olivaceous black, compact, and lacking aerial mycelium. The growth diameters, sugar and salt tolerance, and different cardinal growth temperature of *A. planticola* are shown in Table 3. *A. planticola* can grow at 17–30 °C, and the optimum growth temperature is 28 °C. On MEA supplemented with 10% (*w*/*v*) NaCl, the diameters of *A. planticola* attained 7–8 mm, while its closely related species *A. iranianum* was 5 mm [13]. *A. planticola* can grow on moderate-sugar-level media (PDA), and on both M40Y and M60Y. This species is suitable to grow on M40Y.

Mycelium was dark-pigmented, smooth, thick-walled, branched, and 5.8–10.6 µm (av. = 8.0 µm) wide. Conidiogenous cells were holoblastic, located laterally or terminally, single or in clusters, grey-black to black, and 4.75–2.70 μm (av. = 4.2 μm). Conidia were 5.7–7.7 × 1.5–2.7 μm (av. = 5.2 × 2.8 μm), hyaline, aseptate, smooth-walled, and ellipsoidal to ovoid (Figure 4).

Material examined: —CHINA, Zhejiang, Zhoushan, Miaogen Mountain, from leaf, F, Zixuan, August 2022. (Holotype HMAS 352302; ex-type culture CGMCC 2.7199 = MDSC−10).

Notes—This species is phylogenetically related to *A. thailandense* and *A. castaneae*. These three species can be distinguished by the hyphae color (hyaline or brown in *A. castaneae*, vs. hyaline in *A. thailandense*, vs. dark black in *A. planticola*) [12,19].

#### 3.2.3. *Aureobasidium motuoense* Q.M. Wang, F. Wu & M.M. Wang sp. nov.

Fungal names no: FN 571263.

Etymology: Referring to the location where the ex-type strain originated.

Colonies grew moderately on PDA, MEA, and OA (Figure 2), attaining 36 mm, 42 mm, and 27 mm diameters after 7 days of incubation at 25 °C, respectively. Colonies on PDA were dark brown, with irregular black zones, and sparse aerial mycelium. Colonies on MEA were flat, brownish olivaceous, and white near the margin. Colonies on OA were flat, compact, and pitch black. The growth diameters, sugar and salt tolerance, and different cardinal growth temperature of *A. motuoense* are shown in Table 3. *A. motuoense* can grow at 17–37 °C, and the optimum growth temperature is 30 °C. At 37 °C, the diameters of *A. motuoense* were 7–8 mm, while its relative *A. mangrovei* was 5 mm [15]. *A. motuoense* can tolerate concentrations of up to 15% NaCl. *A. motuoense* grew stronger on moderate-sugar-level media (PDA), and on both M40Y and M60Y. This species is suitable to grow on M40Y.

Mycelium were hyaline to dark brown, smooth-walled, branched, and 1.3–12.7 µm (av. =8.3 µm) wide. Conidiogenous cells were 6.4–12.2 × 3.6–4.5 μm (av. = 8.7× 3.9 μm). Conidia were hyaline, smooth-walled, terminal, and mono- or bipolar budding. Chlamydospores were 12.2–16.2 × 9.6–10.4 μm (av. = 13.6 × 10.0 μm), black, smooth, and globose to elliptic (Figure 5).

Material examined: —CHINA, Tibet, Motuo County, from leaf, W, Guishuang, August 2019. (Holotype HMAS 352304; ex-type culture CGMCC 2.7206 = XZY411−4).

Notes—This species is phylogenetically related to *A. melanogenum* and *A. acericola.* The colonies of *A. melanogenum* on MEA/PDA at 25 °C attained 25 mm diameters after 7 d, appearing smooth and slimy due to abundant sporulation and EPS formation, olive-brown to black in the centre, mustard yellow towards the margin, and at the margin were yellowish white. The colony morphology of *A. acericola* is fast-growing, attaining diameters of 65 mm in 14 days, rapidly turning to olivaceous black, with dark green, irregular margins, covered with slimy masses of conidia, and mycelium immersed or no aerial mycelium [8,10].

#### 3.2.4. *Aureobasidium intercalariosporum* Q.M. Wang, F. Wu & M.M. Wang sp. nov.

Fungal names no: FN 571252

Etymology: Referring to the morphology of intercalary chlamydospores.

Colonies grew moderately on PDA, MEA, and OA (Figure 2), attaining 34 mm, 27 mm, and 28 mm diameters after 7 days of incubation at 25 °C, respectively. Colonies on PDA had an entire margin, and were floccose, greenish black, and white at the edge. Colonies on MEA had an undulate margin, and were flat, brownish olivaceous, grey at the centre, and white near the edge, with sparse aerial mycelium. Colonies on OA were flat, pale, and olivaceous brown to white from the middle to the edge.

The growth diameters, sugar and salt tolerance, and different cardinal growth temperature of *A. intercalariosporum* are shown in Table 3. *A. intercalariosporum* can grow at 17–30 °C, and the optimum growth temperature is 28 °C. On MEA supplemented with 15% (*w*/*v*) NaCl, the diameters of *A. intercalariosporum* attained 9–13 mm, while *A. mangrovei* and *A. pullulans* were 5 mm and 7 mm, respectively [8,16]. Therefore *A. intercalariosporum* grew stronger on concentrations of 15% NaCl than its closely related species. *A. planticola* grew on moderate-sugar-level media (PDA), and on both M40Y and M60Y. This species is suitable to grow on M40Y.

Mycelium was composed of branched, septate hyphae that occurred singly, and were verruculose to smooth, thin-walled, and 1.3–4.8 µm (av. = 3.2 µm) wide. Conidiogenous cells were undifferentiated, smooth, cylindrical, and 4.85–2.68 µm (av. = 3.54 µm) wide. Conidia were 10.5–17.1 × 10.5–12.9 μm (av. = 14.7 × 12.5 μm), smooth-walled hyaline, aseptate, ovoid, and ellipsoidal or elongated ellipsoidal. Chlamydospores were 10.5–17.1 × 10.5–12.9 μm (av. = 14.7 × 12.5 μm), deeply pigmented, smooth, thick walled, and globose to ellipsoidal (Figure 6).

Material examined: —CHINA, Zhejiang, Zhoushan, Miaogen Mountain, from leaf, F, Zixuan, August 2022. (Holotype HMAS 352305; ex-type culture CGMCC 2.7208 = MQL9−100).

Notes—This species is phylogenetically related to *A. pini*, *A. planticol*, *A. khasianum*, *A. leucospermi*, *A. namibiae*, *A. pullulans*, and *A. proteae*.

The main difference between the eight strains is conidia (10.5–17.1 × 10.5–12.9 μm in *A. intercalariosporu*, vs. 4.2–7.4 × 1.7–3.5 μm in *A. planticola*, vs. 6.2–8.5 × 3.6–4.2 μm in *A. pini*, vs. 7–17 × 3.5–7 μm in *A. namibiae*, *vs.* 8–11 × 4–5 μm in *A. leucospermi*, vs. 7.5–16 × 3.5–7 μm in *A. pullulans* [8,14].

### 3.3. Exopolysaccharides Production

Ninety-nine *Aureobasidium* strains were isolated from different leaf and insect samples. Twenty-nine strains among them were selected to evaluate the exopolysaccharides production capacity (Table 3). PTSL19-104 and PTSL20-104 had the highest exopolysaccharide yield, which was 54.8 g/L and 52.33 g/L, respectively. The exopolysaccharides production capacity of *Aureobasidium* strains was different between species. *A. melanogenum* had a strong exopolysaccharide production capacity, with an average exopolysaccharide production of 39.06 g/L. *A. thailandense* was the weakest, with an average exopolysaccharide production of 6.17 g/L. As the interspecific hetero, the intraspecific differences also existed in the production capacity of exopolysaccharides. For example, the lowest sugar production of *A. melanogenum* strain was 1.53 g/L, and the highest sugar production was 54.58 g/L. The strains in the same branch of the phylogenetic tree have the same apparent and microscopic morphology, but the exopolysaccharides yields produced by fermentation are different. For example, the exopolysaccharides yields of four strains among *A. motuoense* are 15.74 g/L, 21.39 g/L, 31.72 g/L, and 37.43 g/L, respectively. In summary, there were differences in exopolysaccharides production capacity among species. Although strain morphology and molecular sequence were consistent within species, there were differences in physiological functions such as exopolysaccharides production capacity.

## 4. Discussion

The species of *Aureobasidium* are widely distributed globally in various habitats, such as house dust, air, tree surfaces (such as needles of *Pinus tabuliformis*, *Acer pseudosieboldianum*, *Bintaro plants*, *Castanea henryi*, and *Castanea mollissima*), plant interiors, seawater, sea ice and glacial meltwaters, water and sediment samples, soil, and subcutaneous phaeohyphomycosis from the US, Canada, Korea, Indonesia, China, the Arctic coast, and Brittany (France) [1,8,10,11,18,21,23]. In this study, many strains were isolated from soil and plant leaves, but strain KCL139 was isolated from the surface of a spittle insect. Yeasts occur commonly on insect species, and the number of insect-related yeasts had increased in the last 10 years reaching a ratio of 7.25% [42,43,44], compared with the report of Boekhout [45]. However, according to our knowledge, *Aureobasidium* species isolated from insects are rare. Our finding not only expands the ecology niches of *Aureobasidium*, but also supports the ‘dispersal–encounter’ hypothesis proposed by Madden et al. [46].

The four new species described in this study have different growth temperatures from each other (Table 2). They can grow at 17 °C, 28 °C and 30 °C; however, only *A. motuoense* can tolerate a high temperature of 37 °C, a feature associated with the climate characteristics of Motuo County where this new species was collected. Motuo County boasts a typical sub-tropical moist climate. The optimum growth temperature for *A. planticola*, *A. intercalariosporum,* and *A. insectorum* is 28 °C, whereas *A. motuoens* is 30 °C. *A. planticola*, *A. intercalariosporum,* and *A. motuoense* fail to grow at 4 °C, whereas *A. insectorum* grows at this temperature normally. Unfortunately, the psychrophilic character of *A. insectorum* cannot be explained based on the present data. *A. planticola*, *A. motuoens,* and *A. insectorum* can grow on MEA supplemented with 15% (*w*/*v*) NaCl, but *A. planticola* can only tolerate concentrations of up to 5% NaCl. All tested *Aureobasidium* species [10], can grow on 10% NaCl MEA, and most of them can tolerate concentrations of 10% NaCl. Only *A. planticola* is intolerant to more than 10% NaCl. *A. planticola* shows good growth on the M40Y medium, while the other strains grow well on the PDA medium.

The study of Zou et al. [47] showed that *A. pullulans* produced polymalic acid by fermentation, and strains ZD-3d, IP-1, Sp. P6, MCW, and ZX-10 produced 57.2 g/L, 57.4 g/L, 91.1 g/L, and 117.4 g/L, respectively, which indicated that different strains could produce different yields of polymalic acid at the same fermentation conditions. In this paper, we found that different strains of the same species, such as *A. insectorum*, *A. leucospermi*, *A. motuoense,* and *A. melanogenum*, produced pullulan ranging from 0.92 to 54.58 g/L (Table 3), which in agreement with the result from Haghighatpanah et al. [37], shows that there were differences in exopolysaccharides production within the same species. Our results also showed that different *Aureobasidium* species have great differences in their exopolysaccharides production capacity. Therefore, we should screen more diverse environments to isolate strains of certain species for comprehensive study in exploring *Aureobasidium* compounds for industrial application.

## 5. Conclusions

In this study, 14 strains of *Aureobasidium* isolated from Tibet, Hebei, and Zhejiang are proposed as four new species based on molecular analysis of their ITS and large ribosomal subunits of D1/D2 domains. The new species are named as *Aureobasidium insectorum* sp. nov., *A. planticola* sp. nov., *A. motuoense* sp. nov., and *A. intercalariosporum* sp. nov. We found differences in interspecific and intraspecific exopolysaccharides production between the species, indicating the diversity in strain-specific exopolysaccharides production. *Aureobasidium* has strong adaptability and wide distribution, and there may be many new *Aureobasidium* taxa in nature, which have the potential to produce new metabolites. Therefore, studies into the species diversity of this genus and the industrial application of its metabolites are of great significance.

## Figures and Tables

**Figure 1 jof-09-00447-f001:**
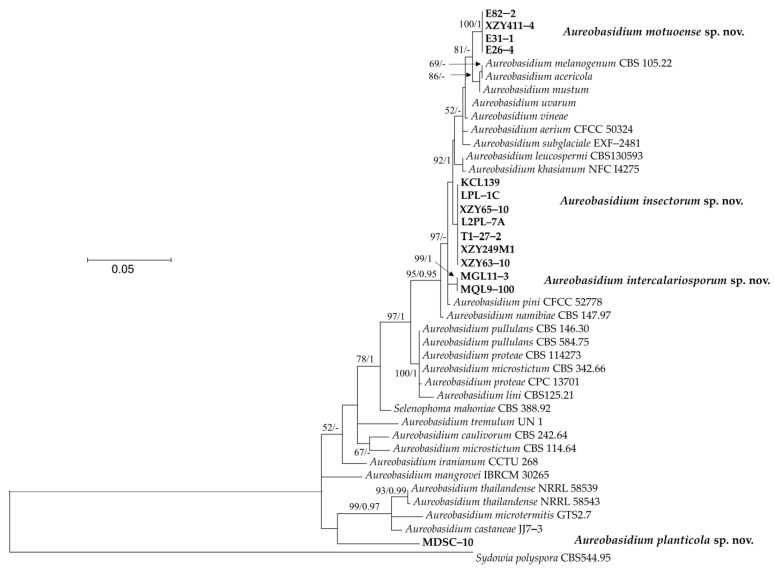
The phylogenetic tree was inferred using the combined sequences of the ITS (including 5.8S rDNA) and LSU rDNA D1/D2 domains, depicting the phylogenetic positions of new taxa (in bold) within *Aureobasidium*. Bootstrap percentages of maximum likelihood analysis over 50% from 1000 bootstrap replicates and Bayesian inference higher than 0.9 (PP > 0.9) are shown on the deep and major branches. Bar = 0.05 substitutions per nucleotide position. Note: -, not supported (BP < 50% or PP < 0.9). The new taxa isolated in this study are shown in bold.

**Figure 2 jof-09-00447-f002:**
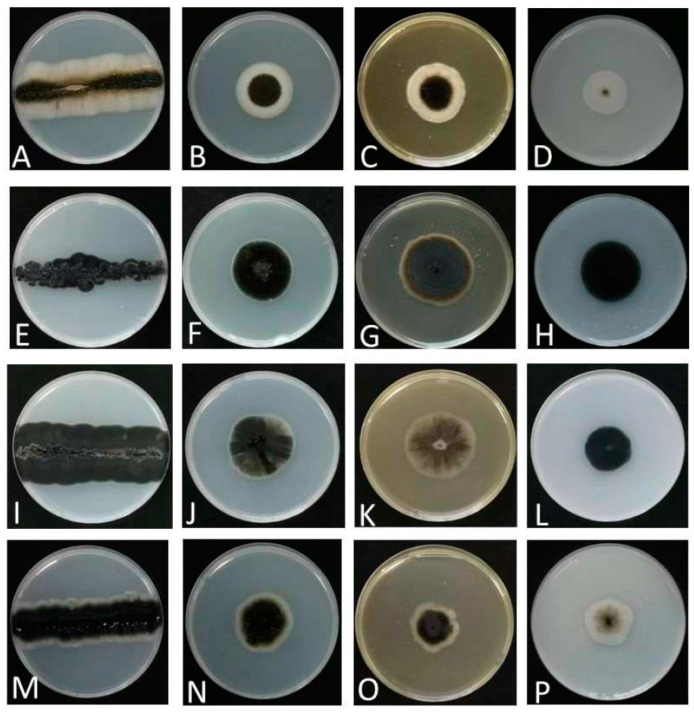
Colony characteristics of analyzed strains. (**A**–**D**): *A. insectorum* KCL139 on PDA, PDA, MEA, and OA, respectively. (**E**–**H**): *A. planticola* MDSC−10 on PDA, PDA, MEA, and OA, respectively. (**I**–**L**): *A. motuoense* XZY411−1 on PDA, PDA, MEA, and OA, respectively. (**M**–**P**): *A. intercalariosporum* MQL9−100 PDA, PDA, MEA, and OA, respectively.

**Figure 3 jof-09-00447-f003:**
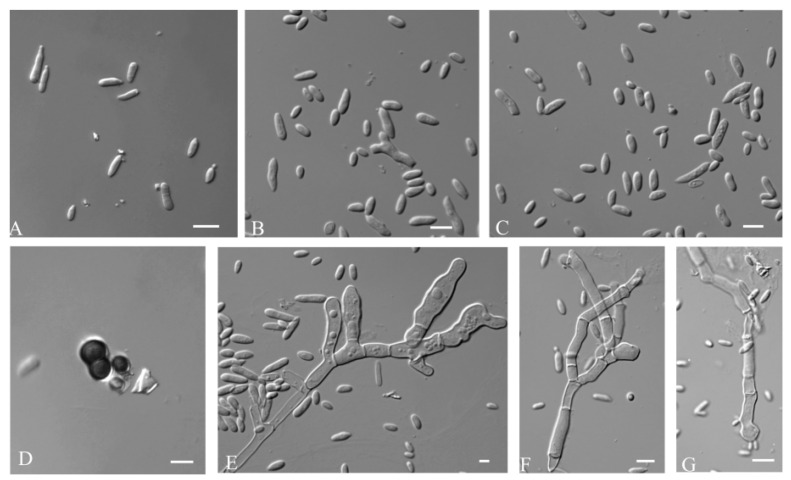
*A. insectorum* (CGMCC 2.7207- ex-type-culture). (**A**–**C**): Conidia. (**D**): Chlamydospores. (**E**–**G**): Conidiogenus cells. Scale bar: (**A**–**G**) = 10 μm.

**Figure 4 jof-09-00447-f004:**
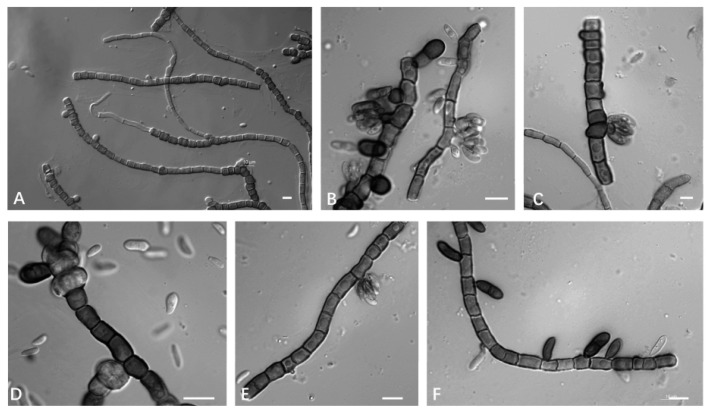
*A. planticola* (CGMCC 2.7199 ex-type culture). (**A**–**F**): Conidiogenous cells and thick-walled hyphae. Scale bar: (**A**–**F**) = 10 μm.

**Figure 5 jof-09-00447-f005:**
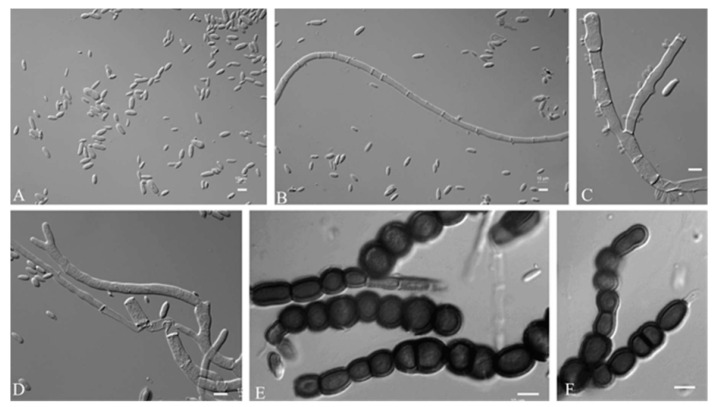
*A. motuoense* (CGMCC 2.7206 ex-type culture). (**A**): Conidia. (**B**–**D**): Hyphae and conidiogenous cells. (**E**,**F**): Chlamydospores. Scale bars: (**A**–**F**) = 10 μm.

**Figure 6 jof-09-00447-f006:**
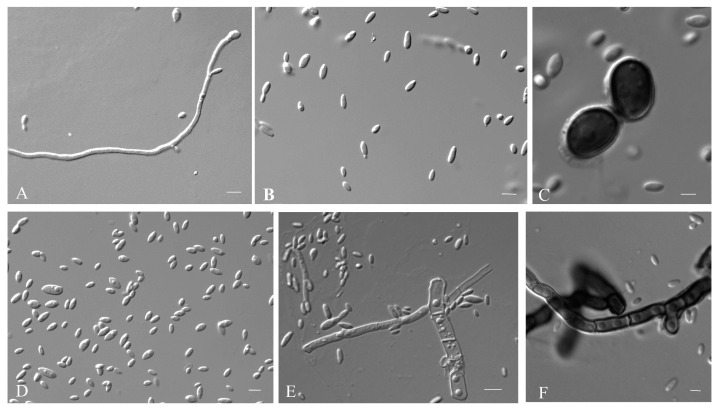
*A. intercalariosporum* (CGMCC 2.7208 ex-type culture). (**A**,**E**): Hyphae and conidiogenus cells. (**B**,**D**): Conidia. (**C**,**F**): Chlamydospores. Scale bars: (**A**–**F**) = 10 μm.

**Table 2 jof-09-00447-t002:** Exopolysaccharides production yield (EPY) of *Aureobasidium*.

Strain	Species	Fermentation Liquid Color	Exopolysaccharides Yield (g/L)	Average Weight (g/L)
PTSL5−5	*A. thailandense*	Light yellow	8.47	6.17
PTSL4−6	*A. thailandense*	Light yellow	8.09	
PTSL5−3	*A. thailandense*	Light yellow	2.71	
PTSL11−5	*A. thailandense*	Light yellow	5.42	
PTSL9−106	*A. melanogenum*	Pink	1.53	39.06
PTSL19−101	*A. melanogenum*	Light yellow	32.53	
PTSL6−101	*A. melanogenum*	Light yellow	34.50	
PTSL19−107	*A. melanogenum*	Yellow	41.84	
PTSL19−104	*A. melanogenum*	Yellow	54.58	
PTSL20−102	*A. melanogenum*	Light yellow	48.13	
PTSL20−104	*A. melanogenum*	Yellow	52.33	
PTSL19−104	*A. melanogenum*	Light yellow	45.71	
PTSL19−104	*A. melanogenum*	Light yellow	45.71	
PTSL17−4	*A. melanogenum*	Light yellow	34.36	
PTSL9−100	*A. melanogenum*	Light yellow	38.45	
LF75−2	*A. leucospermi*	Light yellow	0.92	17.24
SXY35−16	*A. leucospermi*	Light yellow	28.94	
SXY35−15	*A. leucospermi*	Light yellow	23.37	
LF45−2	*A. leucospermi*	Light yellow	15.75	
LPL−7A	*A. insectorum*	Light yellow	27.67	14.7
KCL139	*A. insectorum*	Dark yellow	8.64	
XZY65−10	*A. insectorum*	Dark yellow	7.80	
E26−4	*A. motuoense*	Yellow	15.74	26.57
E31−1	*A. motuoense*	Dark yellow	21.39	
XZY411−4	*A. motuoense*	Dark yellow	31.72	
E82−2	*A. motuoense*	Dark yellow	37.43	
MGL11−3	*A. intercalariosporum*	Light yellow	29.43	31.79
MQL9−100	*A. intercalariosporum*	Light yellow	34.15	
MDSC−10	*A. planticola*	Black	2.10	2.1

**Table 3 jof-09-00447-t003:** Diameter (mm) of four strain colonies under different conditions and on different media. The growth temperature was 25 °C unless noted otherwise and incubation was for 7 d.

	*Aureobasidium planticola*	*Aureobasidium intercalariosporum*	*Aureobasidium motuoense*	*Aureobasidium insectorum*
PDA	22–25	32–36	34–38	32–35
M40Y	33–40	35–40	41–43	32–34
M60Y	27–34	29–34	36–40	25–27
MEA + 5% NaCl	14–15	14–17	13–15	9–12
MEA + 10% NaCl	7–8	8–11	10–11	8–8
MEA + 15% NaCl	0	9–13	5–8	8–9
MEA + 20% NaCl	0	0	0	0
MEA at 4 °C	0	0	0	5–5
MEA at 17 °C	10–13	13–15	8–13	12–15
MEA at 28 °C	27–27	24–25	32–35	28–28
MEA at 30 °C	11–10	13–14	38–44	7–8
MEA at 37 °C	0	0	7–8	0

## Data Availability

All newly generated sequence data are available in NCBI GenBank. The phylogenetic tree and the alignments were deposited in TreeBASE.

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
