# Peer review of "Proposal of Four New Aureobasidium Species for Exopolysaccharide Production"

_jof, 2023, doi:10.3390/jof9040447_

Round 1
Reviewer 1 Report
The current research article reports the topic that impacts the new Aureobasidium species. The authors elucidated the isolation and identification of Aureobasidium, and select some strains to screen for exopolysaccharide production. The overall content of the manuscript is written in an average standard, the alignment of parts is almost appropriate. However, some of the improvement should be concerned.
The analysis method only detects the exopolysaccharide production, not directly observe the pullulan production. Suggest to change the topic according to the product that was measured? Almost the results are taxonomy and phylogenetic analysis, not the pullulan production. So, the topic should be revised?
Line 87-90 The program of PCR of each primer are the same. If the primers are different, it should be carried out in independent PCR reactions?
Line 92 Suggest to put the full name of used media before use the abbreviation.
Line 102-108 “2.4 Phylogenetic Analyses” should put more detail? Used Bayesian inference? How the threes were run?
Line 110 suggest “The starter cultures of the 29 strains were prepared by cultivating the strains in 50 mL of inoculum medium containing 10 g yeast extract, 20 g peptone and 20 g glucose per liter of distilled water at 25 °C for 3 days on a 150-rpm rotary shaker?
Line 112 suggest “5 mL of the starter culture”
Line 114 Please check the format for chemical formular
Line 115 suggest to write the same format with the comment for line 110
Table 1. Why the author used bold text? Suggest to put references in the table and note “Generated sequences in this study are indicated in bold”? Note should be put under the table?
Table 3 Why some strain such as PTSL9-106 (A. melanogenum) and LF75 (A. leucospermi) produced very low yield compared to the other strains from the same species. Is there any discussion about this? The authors should report as the range of the production?
Figure 1 The authors should note about the sequences of the fungal species derived in the current study? Please increase the size of text. It is quite small and difficult to read.
Figure 2 What does the authors means the subscript text “numberT” such as “139T”? Why put in the figure but not put in text or does not have any note?
Figure 3 D-E suggest to use clear photos, cannot see chlamydospore.
Figure 4-6 The resolution of this photo is not good? It is difficult to look for the detail inside the cells. Please look for appropriate example from the other publications.
Line 155 “diam” suggest to use “diameters” same as Line 158.
Line 280 How the authors know that all ninety-nine strains capable of producing exopolysaccharides? Many strains of the genus Aureobasidium can produce but not all strains?
If the author would like to say that the strain produce pullulan, partial purification and characterization by FT-IR spectroscopy should be investigated to confirm that the strains produced pullulan.
Reviewer 2 Report
The topic of this paper is interesting. However, it has some major missing related to the originality of the work, sampling and strains
Abstract section is poor. At the first, authors said they collected 99 strains, and then write about 14 strains?
Authors should revise the material and methodology section of the manuscript. Originality is totally lacking. They did not mention at all about types of leaves and insects from which they isolated the samples. Not write about how many samples they collected and from which they obtained four new species. Which herbarium they deposited the samples.
Whole paper needs appropriate revision.
Round 2
Reviewer 2 Report
Authors responded all the queries satisfactorily. However , there are few spelling mistakes in manuscript that must be corrected before acceptance like
Page 13, line 180, “Mycilium” spelling should be changed to “Mycelium”
Author Response
Point1: Authors responded all the queries satisfactorily. However, there are few spelling mistakes in manuscript that must be corrected before acceptance like
Page 13, line 180, “Mycilium” spelling should be changed to “Mycelium”
Response 1: We have corrected “Mycilium” to “Mycelium” according to the above suggestion, we have also corrected other minor spelling mistakes and grammar in the text, such as “ellipslidal” corrected to “ellipsoidal”, “olivaseous” corrected to “olivaceous”.
We have checked all references are relevant to the contents of the manuscript, and two of them were deleted from the Reference section.